# The Differential Evolution Algorithm for Solving the Problem of Size Selection and Location of Infectious Waste Incinerator

Thitiworada Srisuwandee [ID], Sombat Sindhuchao *[ID] and Thitinon Srisuwandee [ID]

Department of Industrial Engineering, Faculty of Engineering, Ubon Ratchathani University,
Ubon Ratchathani 34190, Thailand
* Correspondence: sombat.s@ubu.ac.th

**Abstract:** The disposal of infectious waste remains one of the most severe medical, social, and environmental problems in almost every country. Choosing the right location and arranging the most suitable transport route is one of the main issues in managing hazardous waste. Identifying a site for the disposal of infectious waste is a complicated process because both tangible and intangible factors must be considered together, and it also depends on various rules and regulations. This research aims to solve the problem of the size selection and location of infectious waste incinerators for 109 community hospitals in the upper part of northeastern Thailand by applying a differential evolution algorithm to solve the problem with the objective of minimizing the total system cost, which consists of the cost of transporting infectious waste, the fixed costs, and the variable cost of operating the infectious waste incinerator. The developed differential evolution produces vectors that differ from the conventional differential evolution. Instead of a single set of vectors, three are created to search for the solution. In addition to solving the problem of the case study, this research conducts numerical experiments with randomly generated data to measure the performance of the differential evolution algorithm. The results show that the proposed algorithm efficiently solves the problem and can find the global optimal solution for the problem studied.

**Keywords:** facility location problem; differential evolution algorithm; infectious waste collection

## 1. Introduction

The disposal of infectious waste remains a severe problem in healthcare waste management in almost every country. However, in the past few years, the level of public concern about healthcare waste management has been increasing worldwide [1]. When mishandled, the collection, transportation, and disposal of infectious waste might cause significant adverse health hazards and environmental impacts [2]. Therefore, infectious waste is hazardous waste that must be a priority in the process of storing, collecting, transporting, and disposing of it properly. It is a critical issue in every country, especially with regard to how to dispose of infectious waste because the disposal of the infectious waste directly affects the environment and people. Infectious waste occurs in medical diagnosis and treatment, immunization, and disease experiments and includes cotton buds, gauze, needles, blades, rubber tubes, excretions, bodily fluid flow from the patient, etc. In 2020, Thailand had a total of 47,962 tons of infectious waste from government hospitals, private hospitals, private clinics, veterinary hospitals, and dangerous infection laboratories. From the report on the situation of infectious waste in Thailand from 2015 to 2020, the amount of infectious waste tended to increase continuously, as shown in Figure 1.

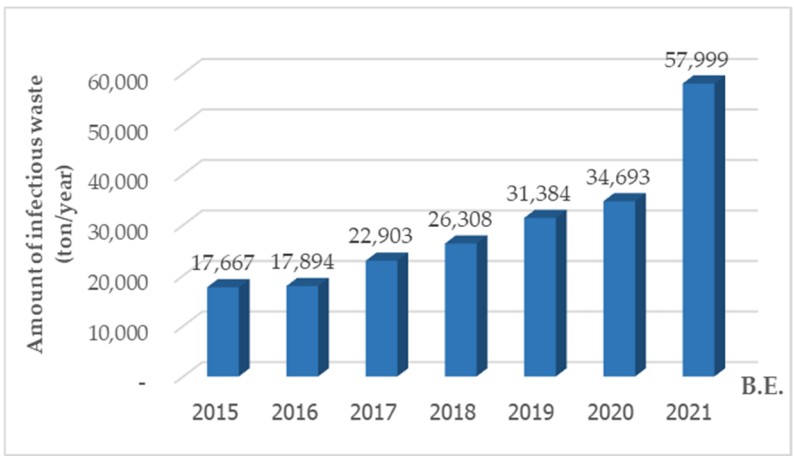

**Figure 1.** The amount of infectious waste that occurred from 2015 to 2021 [3].

Chummuel et al. [4] surveyed infectious waste incinerators in the upper part of the northeastern region for proper management. It was found that every community hospital had an incinerator for infectious waste disposal. However, only 22.09% of the incinerators were in regular use, and the rest were out of order due to the lack of spare parts and proper maintenance. Therefore, the community hospitals whose incinerators did not work would send infectious waste to a private company for disposal; the company was located in the central part of Thailand. As a result, these community hospitals had to pay the high cost of transporting infectious waste to the waste disposal site. Moreover, the infectious waste collection was slow because the burning capacity of the existing incinerator was not enough to meet the demand. As a result, the Thai government released a policy to promote the establishment of new waste disposal centers in potential hospital areas to fix such problems and increase efficiency in the removal of infected garbage. These disposal facilities had to be able to serve nearby hospitals and at the same time minimize the economic, environmental, health, and social factors. New disposal facilities needed to be planned along with proper transport routing to have the lowest transport routing costs for maximum benefit. Therefore, choosing a site for the disposal of infectious waste was a complicated and challenging process because it had to comply with the regulatory requirements of each country and had to reduce the social, environmental, and economic impacts at the same time [5].

The upper part of northeastern Thailand has 109 community hospitals under the Ministry of Public Health. These community hospitals often encounter problems due to their remoteness from the existing external waste disposal agencies. Choosing the right size of the new incinerator and its suitable location, in this case, is a complex problem that is difficult to solve using existing techniques alone because there are relevant factors to consider, including infrastructural, geological, environmental, social, and exact cost factors. All the factors must be considered simultaneously for maximum benefit. To address the above problem, this study focuses on the problem of selecting the size and location of the hospital incinerator for infectious waste disposal sites located in northeastern Thailand. The objective is to obtain the lowest total cost of infectious waste transportation and operating costs combined with the operating costs of the infectious waste incinerator. Sresanpila and Sindhuchao [6] proposed a mathematical model and particle swarm optimization (PSO) to solve the problem of the size selection and location of infectious waste incinerators. The objective function and some constraints of the mathematical model are non-linear. Later, Reference [7] developed an iterated local search (ILS) to solve this problem. Using mathematical programming, both [6,7] could not find the optimal solution due to the non-linear components. As a result, this research proposes an adjustment of the mathematical model from non-linear to linear without changing the nature of the problem. Due to the complex nature of the problem and its model, the problem is NP-hard. If the problem is very large, solving problems with mathematical modeling will take quite a long time, and

the results may not be found. Therefore, the authors propose a method of evolution using differential evolution (DE).

Our main contributions are as follows:

1.  We develop a differential evolution algorithm that can efficiently solve the problem.
2.  We propose an adjustment of the mathematical model from non-linear to linear without changing the nature of the problem, which Lingo Solver can solve to optimality.

The remainder of the article is organized as follows: Section 2 presents the related literature; Section 3 presents the details of the original and modified mathematical models. The differential evolution algorithm (DE) is proposed in Section 4. The experimental results are discussed in Section 5, and finally, the conclusion of this research is presented in Section 6.

## 2. Related Literature

The Facility Location Problem (FLP) has been studied for a long time and was first studied by Pierre de Fermat, Evagelistica Torricelli, and Battista Cavallieri [8]. The original FLP was a problem with a single objective or goal. The objectives often consider location issues because location decisions directly affect the organization's operational and logistical decisions, which are related to service capability [9]. In general, the FLP is used to determine the number, the size, and the locations of the facilities and to allocate services from these locations to customers both within and outside the organization to minimize the transportation costs, distance, or time required to deliver the goods or services [10]. Alumur and Kara [11] propose a mathematical model to solve the problem of hazardous waste site routing by selecting a site for hazardous waste storage and disposal. The use of technology is considered to select hazardous waste disposal sites and to arrange the suitable transport routes in order to reduce the total costs and the risks of hazardous waste transportation. Aboutahoun [12] proposes a mathematical model for selecting hazardous waste disposal or storage sites and transportation routes by mainly considering the cost of the damage caused by accidents. The mathematical model is developed in conjunction with Floyd Warshall's algorithm to provide the shortest transport distance and the lowest risk of accidents. Wichapa and Khokhajaikiat [13] propose a solution to the site routing problem of infectious waste by selecting suitable areas to establish infectious waste disposal facilities for community hospitals in the upper part of northeastern Thailand. A hybrid fuzzy goal programming model is used, which combined fuzzy analysis, a hierarchy process, and fuzzy goal programming. Later, Reference [14] determines the shortest distance for collecting infectious waste using a hybrid genetic algorithm that mixes genetic algorithms and local search. These include 2-Opt-move, Insertion-move, and λ-interchange-move, resulting in the lowest total cost. Suksee and Sindhuchao [15] suggest heuristic ideas to solve the problem of waste incineration site selection and the routing of infectious waste collection vehicles for hospitals in northeastern Thailand based on the principles of the greedy randomized adaptive search procedure (GRASP) and the adaptive large neighborhood search (ALNS). A simulation optimization approach is proposed by [16]. It integrates a system dynamics simulation model (SDSM) with a multi-period capacitated facility location problem (CFLP) as a decision support tool for future APL implementations. In addition, a Monte Carlo simulation is applied to estimate the costs and reliability level with random demands. Němec et al. [17] present the possibilities in solving the weighted multi-facility location problem (MFLP) and its related optimization tasks using a widely available office software—MS Excel with the Solver add-in. The result shows that this widely available office software is practical when solving even relatively complex optimization tasks, with sufficient quality for many real-world applications. Yu et al. [18] propose the regional location routing problem (RLRP) model and the multi-depot regional location routing problem (MRLRP) model, which are extensions of the location routing problem (LRP), to provide a better municipal waste collection process. In addition, hybrid genetic algorithm-simulated annealing is applied to determine the depot locations in each region and the vehicles' routes

for collecting waste to fulfill the inter-regional independent needs at a minimum total cost. The results show that the proposed method efficiently solves the RLRP and MRLRP.

The differential evolution algorithm is presented by [19]. This method has a process of finding solutions which is similar to that of the genetic algorithm (GA). The improvement of the differential evolution algorithm to increase the solution's efficiency is developed by [20], who present five mutation strategies called DE/rand/1, DE/best/1, DE/rand-to-best/1, DE/best/2, and DE/rand/2. Chiang et al. [21] propose a method for modifying coordinates by applying the principle of 2–Opt exchange. Pitakaso et al. [22] introduce 1-point and 2-point exponential exchanges for exchanging coordinates. The binomial exchange of coordinates slowly leads to good solutions, while the 1-point exponential coordinate exchange moves towards a good solution quickly. In the early stages of the differential evolution algorithm, the 2-point exponential coordinate exchange is slow to lead to good solutions, but after a period of processing, good solutions can be obtained quickly. In addition, both the 1-point and the 2-point exponential coordinate exchanges provide a similar result. Zhu et al. [23] present a method for vector scaling. If a search cannot find a better solution, a new vector will be added. The values in the coordinates of the newly created vector are randomly numbered between 0 and 1. Then, the procedures of the differential evolution algorithm are repeated. If no better solution is found, the population size keeps increasing until it reaches the maximum population limit. Sethanan and Pitakaso [24] develop the differential evolution algorithm with two types of local search for solving the general assignment problem. Type 1 performs a local search on all trial vectors. Type 2 performs a local search on some vectors only—10% of the best vectors and 10% of the remaining vectors—in order to reduce the computational time. The results show that the solutions can be improved with less computational time. In addition to improving the methodology of the differential evolution algorithm, the hybrid approaches between the differential evolution and some metaheuristics are also introduced by [25–30]. Epitropakis et al. [25], Miranda and Alves [26], Sedki and Ouazar [27], Thongdee and Pitakaso [28] present the differential evolution algorithm with the PSO, while [29,30] propose the hybrid differential evolution with the simulated annealing (SA). These hybrid approaches perform the common DE more efficiently.

The studies discussed above show that global search metaheuristics (e.g., DE, GA, PSO, ALNS, and SA) are effective in solving optimization problems such as the location problem. Due to the importance of the environmental and economic impact, the cost of transporting infectious waste and the cost of incinerating infectious waste are considered. As a result, the objective of the problem studied is to minimize total energy consumption while considering the frequency of traveling to collect infectious waste and the size of the infectious waste incinerators. In order to solve this problem, a mathematical model has been adjusted, and DE is introduced to further the search efficiency of the solution.

## 3. Problem Formulation

### 3.1. Problem Description

The problem of the size selection and location of infectious waste incinerators for 109 community hospitals in the upper part of northeastern Thailand was studied. Figure 2 shows the locations of all 109 community hospitals whose total amount of infectious waste was approximately 104,487 kg/month. Each community hospital can be selected as the location of the infectious waste disposal facility where only one incinerator can be operated. There are three types of incinerators, with different waste-burning capacities: 100, 300, and 600 kg/h. Each hospital can obtain the service of infectious waste disposal from only one facility. The incinerator can be operated continuously for a month after a 6 h warm-up time. The transportation of infectious waste from the hospital to the incinerator location is by direct shipping. A vehicle is dispatched from the infectious waste disposal facility to collect the infectious waste at a particular community hospital, and then, it returns to the facility. The community hospitals may have different frequencies of infectious waste disposal due to different waste management policies. The solution is to determine where the infectious

waste disposal facilities should be located and which type of incinerator should be utilized at each facility, as well as to assign the community hospitals to each facility in order to minimize the total system cost, which comprises the transportation cost and the fixed and variable operating costs of the incinerator.

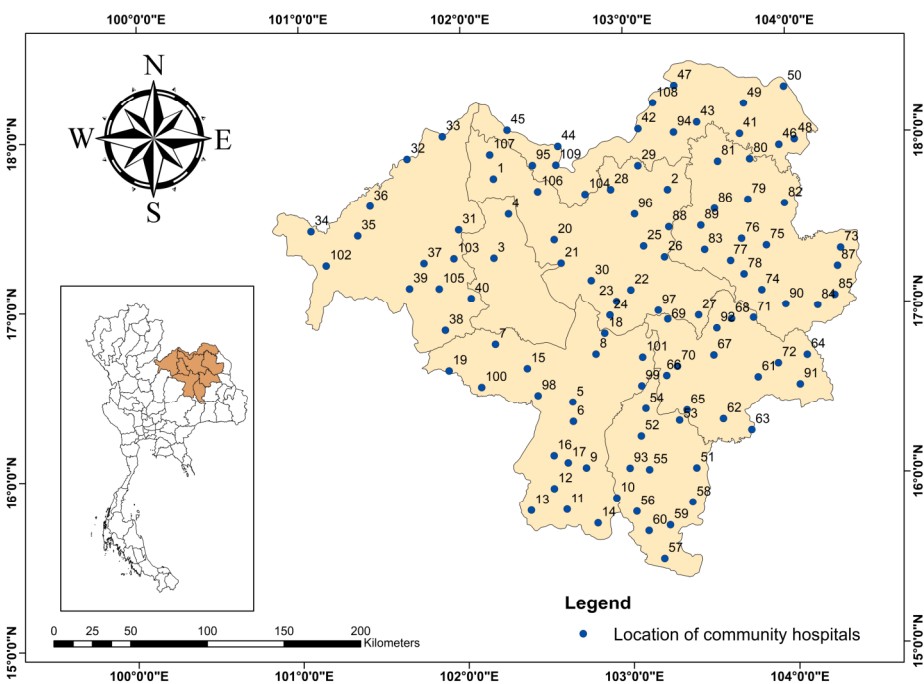

**Figure 2.** Location of community hospitals in the upper part of northeastern Thailand.

### 3.1.1. Data Collection

The data were collected on the infectious waste from 109 community hospitals in the upper part of northeastern Thailand, comprising nine provinces: Khon Kaen, Kalasin, Mahasarakham, Sakon Nakhon, Udon Thani, Nong Khai, Nong Bua Lamphu, Loei, and Bueng Kan [6,31]. The courtesy information was from private companies and community hospitals and included the amount of infectious waste received for disposal in each hospital, the frequency of the receiving of infectious waste for disposal, the service charge rate/time, and the shipping cost/km. Therefore, the preliminary data could be collected as follows.

### 3.1.2. Information about Infectious Waste Incinerators

Incinerators used to dispose of infectious waste come in many forms, divided according to the ability to burn infectious waste (kg/time unit). In this study, only three types of incinerators were considered: the incinerators capable of incinerating infectious waste of 100, 300, and 600 kg/h. It is the model that private service providers recommend or currently use. The information about the price, the maximum burning rate, and the service life of each type of incinerator is shown in Table 1.

**Table 1.** Information on the infectious waste incinerators considered.

| Information | Infectious Waste Incinerator | | |
|---|---|---|---|
| | **Type 1** | **Type 2** | **Type 3** |
| Price (THB) | 1,995,000 | 3,745,000 | 10,165,000 |
| Maximum Burning Rate (kg/hour) | 100 | 300 | 600 |
| Service Life (Years) | 10 | 10 | 10 |

The costs considered consist of two essential parts: the cost of transporting the infectious waste and the cost of incinerating the infectious waste. The shipping costs depend on the distance of the transport and the cost per unit distance. The cost of incinerating infectious waste consists of several parts: incinerator maintenance fee, utility bills, fuel cost for burning, employee wages, and depreciation. These costs are based on the research data from [6,31]. The cost of the incineration of infectious waste by incinerator type is shown in Table 2.

**Table 2.** The cost of incineration of infectious waste/hour.

| Characteristics of Each Type of Infectious Waste Incinerator | Type 1 | Type 2 | Type 3 |
|---|---|---|---|
| Incinerator Maintenance Fee (THB/Hour) | 53 | 90 | 128 |
| Utility Bills (THB/Hour) | 107 | 135 | 183 |
| Fuel Cost for Burning (THB/Hour) | 210 | 329 | 607 |
| Cost of Incineration of Infectious Waste (THB/Hour) | 370 | 554 | 918 |
| Depreciation (THB/Month) * | 16,397 | 30,781 | 83,548 |
| Employee Wages (THB/Month) ** | 31,500 | 31,500 | 31,500 |
| Fixed Cost of Incinerator Operation (THB/Month) | 47,897 | 62,281 | 115,048 |

* Depreciation is calculated from incinerator price: THB/lifetime (days). ** Employee wages are calculated from 350 THB/day/person (three employees).

### 3.1.3. Transport Information for Infectious Waste

1. The distance to each hospital uses the latitude and longitude of the hospital location, to be processed via Google Map. The distance is in kilometers.
2. The vehicles used to collect infectious waste from each hospital use six-wheel trucks. The average speed of the truck is 60 km/h. The work time of the garbage collector is 12 h/day (including a 1 h break).
3. The cost of transporting infectious waste is 5 THB/km, and the average weight of the infectious waste is 60 kg/bin.
4. The average waste collection time is 2 min/bin, and the average service time is 10 min/time at each community hospital.

### 3.2. The Mathematical Model for the Problem of Size Selection and Location of the Infectious Waste Incinerators

The mathematical model of the problem of the size selection and location of the infectious waste incinerators formulated by [6] has been studied. The original objective function and some of the constraints are non-linear. As a result, it is difficult to solve for the global optimal solution. In this research, the non-linear terms have been adjusted to become linear, which makes it easier to obtain the global optimal solution. The details of the mathematical model and its improvement are as follows.

**Indices**

$i$ Sequence of community hospitals $i = 1, 2, \ldots, I$;

$j$ Sequence of positions that can open the incinerators for infectious waste $j = 1, 2, \ldots, J$;

$k$ Sequence of the infectious waste incinerator model $k = 1, 2, \ldots, K$.

**Parameters**

$C_{ij}$ Transportation distance from the hospital $i$ to the location of the infectious waste incinerator $j$;

$P_k$ Fixed cost of operating the infectious waste incinerator $k$;

$D_i$ The average amount of infectious waste generated each month in community hospitals;

$A$ Cost of transporting infectious waste;

$F_i$ Frequency of traveling to collect infectious waste from hospital $i$ each month;

$B_k$ The maximum burning rate per hour of the incinerator $k$;
$O_k$ Cost of operating the incinerator for infectious waste $k$.
**Decision Variables**

$$X_{ij} = \begin{cases} 1, & \text{if the infectious waste of the community hospital } i \text{ is transported to location } j \\ 0, & \text{Otherwise} \end{cases}$$

$$Y_j = \begin{cases} 1, & \text{If the infectious waste incinerator is open at the location } j \\ 0, & \text{Otherwise} \end{cases}$$

$$S_{kj} = \begin{cases} 1, & \text{if the incinerator } k \text{ is used at location } j \\ 0, & \text{Otherwise} \end{cases}$$

$H_{kj}$ Number of hours per month that the incinerator $k$ is operated at the location $j$

$$Min\ Z = \sum_{i=1}^{I}\sum_{j=1}^{J} AF_i C_{ij} X_{ij} + \sum_{j=1}^{J}\sum_{k=1}^{K} P_k S_{kj} + \sum_{j=1}^{J}\sum_{k=1}^{K} H_{kj} O_k S_{kj} \tag{1}$$

$$Min\ Z = \sum_{i=1}^{I}\sum_{j=1}^{J} AF_i C_{ij} X_{ij} + \sum_{j=1}^{J}\sum_{k=1}^{K} P_k S_{kj} + \sum_{j=1}^{J}\sum_{k=1}^{K} H_{kj} O_k + \sum_{j=1}^{J}\sum_{k=1}^{K} (6) S_{kj} O_k \tag{2}$$

The objective function (1) is to minimize the total system cost which is the sum of the cost of transporting infectious waste and the fixed cost of the operation combined with the variable cost of operating the infectious waste incinerator. The term $\sum_{j=1}^{J}\sum_{k=1}^{K} H_{kj} O_k S_{kj}$ is non-linear due to the multiplication of two decision variables, $H_{kj}$ and $S_{kj}$. For this term, $H_{kj}$ is the number of operating hours of the incinerator per month, including the 6 h warm up time. To adjust this term to be linear, let $H_{kj}$ be the number of waste-burning hours only, not including the 6 h warm up time. However, after adjusting the equation, $H_{kj}$, the warm-up hours of the infectious waste incinerator are not included. Then, $\sum_{j=1}^{J}\sum_{k=1}^{K} H_{kj} O_k S_{kj}$ can be divided into two terms: $\sum_{j=1}^{J}\sum_{k=1}^{K} H_{kj} O_k + \sum_{j=1}^{J}\sum_{k=1}^{K} (6) S_{kj} O_k$, as shown in (2). The first term is the cost of burning the infectious waste of the incinerator, and the second term is the cost of warming up the incinerator.

Constraint (3) is a conditional equation forcing each community hospital $i$ to obtain service from only one infectious waste disposal facility.

$$\sum_{j=1}^{J} X_{ij} = 1, \forall i, i = 1, 2, \ldots, I \tag{3}$$

Constraint (4) stipulates that the community hospital selected as an infectious waste disposal facility must use only one type of incinerator.

$$\sum_{k=1}^{K} S_{kj} = Y_j, \forall j, j = 1, 2, \ldots, J \tag{4}$$

Constraint (5) defines that each community hospital can receive infectious waste disposal services only from the facilities that are in operation.

$$X_{ij} \leq Y_j, \forall i, j, i = 1, 2, \ldots, I \text{ and } j = 1, 2, \ldots, J \tag{5}$$

Constraint (6) indicates that the infectious waste incinerator of the facility is required to warm up for 6 h before starting an incineration process, and in each month, the number of incinerating hours must be sufficient to eliminate all the infectious waste collected

from the community hospitals assigned to the facility. Due to the fact that the term of $\sum_{k=1}^{K} B_k S_{kj} \left( H_{kj} - 6 \right)$ is in a non-linear form, it would take a lot of time and effort to obtain the optimal solution. The problem can be eased by transforming this term to become linear. To do so, $H_{kj}$ is given a new meaning, which is the total time of incineration without the warm-up time. Then, constraint (6) can be replaced by constraint (7).

$$\sum_{k=1}^{K} B_k S_{kj} \left( H_{kj} - 6 \right) \geq \sum_{i=1}^{I} D_i X_{ij}, \ \forall j, \ j = 1, 2, \ldots, J \tag{6}$$

$$\sum_{k=1}^{K} B_k H_{kj} \geq \sum_{i=1}^{I} D_i X_{ij}, \ \forall j, \ j = 1, 2, \ldots, J \tag{7}$$

Constraint (8) gives the limit of the number of hours that the infectious waste incinerator $k$ can be operated continuously at location $j$, which must not be more than one month or 30 days, including the 6 h warm-up time. To cooperate with constraint (7), constraint (9) replaces constraint (8) by excluding the 6 h warm-up time from the upper bound.

$$0 \leq H_{kj} \leq (24)(30) S_{kj}, \ \forall k, j, \ k = 1, 2, \ldots, K \ and \ j = 1, 2, \ldots, J \tag{8}$$

$$0 \leq H_{kj} \leq ((24)(30) - 6) S_{kj}, \ \forall k, j, \ k = 1, 2, \ldots, K \ and \ j = 1, 2, \ldots, J \tag{9}$$

Constraint (10) states that if location $j$ is selected as the infectious waste disposal facility, there must be at least one community hospital using its services. Constraints (11), (12), and (13) define the binary decision variables.

$$\sum_{i=1}^{I} X_{ij} = Y_j, \ \forall j, \ j = 1, 2, \ldots, J \tag{10}$$

$$X_{ij} \in \{0, 1\}, \ \forall i, j, \ i = 1, 2, \ldots, I \ and \ j = 1, 2, \ldots, J \tag{11}$$

$$Y_j \in \{0, 1\}, \ \forall j, \ j = 1, 2, \ldots, J \tag{12}$$

$$S_{kj} \in \{0, 1\}, \ \forall k, j, \ k = 1, 2, \ldots, K \ and \ j = 1, 2, \ldots, J \tag{13}$$

## 4. Differential Evolution (DE) Algorithm

The differential evolution (DE) algorithm is metaheuristic one and has an iteration process that has been used extensively for solving difficult optimization problems. It is a population-based method inspired by biological evolution, including mutation, recombination, and selection. DE uses vectors to generate solutions. New offspring solutions can be produced through mutation, recombination, and selection. To apply DE for solving a particular problem, the vectors must be well designed to come up with the solution. The case study is a problem in which there are many decision variables: the number of incinerators, the incinerator sizes, the incinerator locations, and the assignment of the community hospitals to each location. Therefore, the solution vectors are designed into three sets (X1, X2, X3) for the easy conversion of the answer from the vectors. The details of the initialization of the vectors, mutation, recombination, and selection can be explained as follows.

### 4.1. Initialization of Vectors

To solve the problem of selecting the size and location of the infectious waste incinerator, a certain number of vectors called "Target Vector" are created in the first iteration. Each target vector will consist of three sets of vectors. The first vector (X1) is used to find the number of infectious waste disposal facilities that are to open. The second vector (X2) determines the locations of the incinerators or facilities and the assignment of the community hospitals to each facility. Finally, the third vector (X3) is used to find the type

of incinerator that should be operated at each location. The vectors X1, X2, and X3 have 109 coordinates, which is equal to the number of community hospitals. Coordinate 1 means community hospital 1, coordinate 2 means community hospital 2, and so on. For each coordinate of vector X3, there are three sub-coordinates equal to the number of incinerator types considered. Sub-coordinate 1 means incinerator type 1. The algorithm begins with the randomization of a real number between 0 and 1, which is assigned to each coordinate of vectors X1 and X2 and to each sub-coordinate of vector X3, as shown in Table 3. Then, the initial solution to the problem can be obtained from the vector values.

**Table 3.** Examples of values in the coordinates of one target vector consisting of three vectors.

| Vector Sets | Coordinates (Community Hospitals) | | | | | |
|---|---|---|---|---|---|---|
| | 1 | 2 | 3 | . . . | 108 | 109 |
| X1 | 0.78 | 0.45 | 0.2 | . . . | 0.52 | 0.15 |
| X2 | 0.8 | 0.91 | 0.85 | . . . | 0.81 | 0.35 |
| X3 | 0.1, 0.85, 0.43 | 0.58, 0.24, 0.9 | 0.33, 0.18, 0.75 | . . . | 0.65, 0.2, 0.88 | 0.45, 0.31, 0.9 |

To obtain an initial solution, the vector X1 is first considered in order to identify which coordinate of the vector X1 has the greatest random number value. The number of infectious waste disposal facilities to open is equal to the order of the coordinate that has the largest random number. For example, it is supposed that the random number of the fifth coordinate is the largest; so, the solution is to set up five infectious waste disposal facilities. Then, the vector X2 is considered in order to find which community hospitals will be the locations of the facilities. To do so, the location candidate list is formed by sorting the coordinates of the vector X2 in descending order of the random numbers. Due to opening five facilities, the first five orders of the community hospitals in the location candidate list will be the locations of the facilities. For example, it is assumed that the first five orders in the location candidate list are the coordinates 52, 25, 15, 2, and 33. As a result, community hospitals 52, 25, 15, 2, and 33 will be selected as the locations of the infectious waste disposal facilities. To determine which type of incinerator will be used at each location, the vector X3 is considered. For each coordinate of the selected locations, the sub-coordinate with the largest random number identifies the type of incinerator. For example, if coordinate 52 has sub-coordinates with random numbers equal to 0.2, 0.5, and 0.7, then the sub-coordinate 3 has the largest random number (0.7). This means that incinerator type 3 will be chosen to be operated at the facility located at community hospital 52. Returning to the location candidate list, an assignment of community hospitals to each facility will be managed. Starting with the first order in the location candidate list, each community hospital is assigned to the nearest facility under all the constraints, such as the burning capacity of the incinerator. The assignment process continues until all community hospitals have been assigned to the facilities. From the previous example, starting with community hospital 52 in the first order of the list, it will be assigned to facility 1, which is located at the location of community hospital 52 because this hospital is closest to facility 1 with the 600 kg/hour burning capacity of incinerator type 3; all the monthly infectious waste of community hospital 52, combined with one of the other community hospitals previously assigned to facility 1, can be eliminated. Once the assignment of the community hospitals to the facilities is completed, the fitness function value can be computed. In this case, the fitness function is the objective function (2). A target vector provides one solution. The number of target vectors that should be created as a population depends on the problem size. In general, a large population can find a better solution than the small one. The vector giving the best fitness value or the lowest total system cost is retained, and it will be replaced if a better solution is found.

### 4.2. Mutation

The DE mutation strategy presented in this research is DE/best/2, where the number of difference vectors involved is two, and the best target vector is considered [20]. After generating a number of target vectors (population size) and evaluating their fitness values, the vector with the best fitness value will be used to construct all the mutant vectors. The variables related to the mutation process are shown below.

$X1_{i,G}, X2_{i,G}, X3_{i,G}$   Target vector $i$ in iteration G.

$V1_{i,G}, V2_{i,G}, V3_{i,G}$   Mutant vector $i$ in iteration G.

$X1_{best,G}, X2_{best,G}, X3_{best.G}$ Vector giving the lowest fitness value in iteration G.

$X1_{r1,G}, X1_{r2,G}, X1_{r3,G} X1_{r4,G}$  Four random vectors of vector set 1 in iteration G.

$X2_{r1,G}, X2_{r2,G}, X2_{r3,G} X2_{r4,G}$  Four random vectors of vector set 2 in iteration G.

$X3_{r1,G}, X3_{r2,G}, X3_{r3,G} X3_{r4,G}$  Four random vectors of vector set 3 in iteration G.

$F$ Scaling factor (constant real number equal to 2 in this case).

In each iteration, the mutant vector $i$ can be obtained using Equations (14)–(16). As seen, to obtain a mutant vector $i$ of any given vector set (1, 2 or 3) in any given iteration, the best target vector of that set in that iteration will be used along with four randomly chosen target vectors of that set in that iteration.

$$V1_{i,G} = X1_{best,G} + F(X1_{r1,G} - X1_{r2,G}) + F(X1_{r3,G} - X1_{r4,G}) \tag{14}$$

$$V2_{i,G} = X2_{best,G} + F(X2_{r1,G} - X2_{r2,G}) + F(X2_{r3,G} - X2_{r4,G}) \tag{15}$$

$$V3_{i,G} = X3_{best,G} + F(X3_{r1,G} - X3_{r2,G}) + F(X3_{r3,G} - X3_{r4,G}) \tag{16}$$

After the mutation process, the mutant vectors are obtained with different values of coordinates from those of the target vectors. Therefore, new solutions (offspring) to the problem are generated and can be determined with the same schemes mentioned in Section 4.1. Finally, the best solution is updated.

### 4.3. Recombination

In this recombination process, a trial vector (offspring) is produced in a probabilistic manner using the values of the coordinates of both the mutant and the target vectors. With different values of coordinates from the parents (target and mutant vectors), the variety of solutions to the problem can be obtained. How to obtain the values of the coordinates of the trial vectors can be explained by Equations (17)–(19). To compute a value of the coordinate of any trial vector, a real number between 0 and 1 ($rand(j)$ $and$ $rand(j,k)$) will be randomly selected. If the random number is less than or equal to the coordinate exchange rate (CR), the value in this coordinate position of the trial vector is equal to the value in the same coordinate position of the mutant vector. In the other case, it is equal to that of the target vector. The same condition holds for all the components of the trial vectors.

$$U1_{ij,G} = \begin{cases} V1_{ij,G}, if(rand(j) \le CR) \\ X1_{ij,G}, Otherwise \end{cases} \tag{17}$$

$$U2_{ij,G} = \begin{cases} V2_{ij,G}, if(rand(j) \le CR) \\ X2_{ij,G}, Otherwise \end{cases} \tag{18}$$

$$U3_{ijk,G} = \begin{cases} V3_{ijk,G}, if(rand(j,k) \le CR) \\ X3_{ijk,G}, Otherwise \end{cases} \tag{19}$$

$U1_{ij,G}, U2_{ij,G}$: Value of coordinate $j$ of trial vector $i$ of vector sets 1 and 2 in iteration $G$.

$U3_{ij,G}$: Value of sub-coordinate $k$ of coordinate $j$ of trial vector $i$ of vector set 3 in iteration $G$.

$V1_{ij,G}$, $V2_{ij,G}$: Value of coordinate $j$ of mutant vector $i$ of vector sets 1 and 2 in iteration $G$.

$V3_{ij,G}$: Value of sub-coordinate $k$ of coordinate $j$ of mutant vector $i$ of vector set 3 in iteration $G$.

$X1_{ij,G}$, $X2_{ij,G}$: Value of coordinate $j$ of target vector $i$ of vector sets 1 and 2 in iteration $G$.

$X3_{ij,G}$: Value of sub-coordinate $k$ of coordinate $j$ of target vector $i$ of vector set 3 in iteration $G$.

$rand(j)$: Random number in coordinate $j$.

$rand(j, k)$: Random number in coordinate $j$ and in sub-coordinate $k$.

$CR$: Coordinate exchange rate.

Equations (17)–(19) are used to produce the trial vectors of sets 1, 2, and 3, respectively. Table 4 demonstrates the construction of a trial vector of set 1 using the coordinate exchange rate of 0.8. For coordinate 1, the values of coordinate 1 of the target vector and the mutant vector are 0.78 and 0.45, respectively. A random number of 0.56 is chosen, and it is less than the CR. As a result, from Equation (17), the value of coordinate 1 of the trial vector will be 0.45. The same comparison scheme is applied for the remaining components of this trial vector. Once all the trial vectors in a given iteration have been created, a new solution can be interpreted from each trial vector, and the best solution is updated again.

**Table 4.** Examples of trial vector determinations of the first set of vectors at CR = 0.8.

| Vector Set | Coordinate | | | | | |
|:---:|:---:|:---:|:---:|:---:|:---:|:---:|
| | **1** | **2** | **3** | **. . .** | **108** | **109** |
| X1 | 0.78 | **0.45** | 0.2 | . . . | 0.52 | **0.15** |
| V1 | **0.45** | 0.11 | **0.95** | . . . | **1.25** | 0.84 |
| Random | 0.56 | 0.9 | 0.73 | . . . | 0.15 | 0.98 |
| U1 | 0.45 | 0.45 | 0.95 | . . . | 1.25 | 0.15 |

*4.4. Selection*

Selection is the procedure for selecting the target vectors for the next iteration, which can be obtained from Equation (20). The target vector $i$ for the next iteration $(X_{i,G+1})$ will be either the trial vector $i$ of the current iteration $(U_{i,G})$ or the target vector $i$ of the current iteration $(X_{i,G})$. If the trial vector has a better fitness value than that of the target vector, the target vector for the next iteration will be the trial vector of the current iteration. In the other case, the target vector for the next iteration will be the target vector of the current iteration.

$$X_{i,G+1} = \begin{cases} U_{i,G}, if f(U_{i,G}) < f(X_{i,G}) \\ X_{i,G}, Otherwise \end{cases} \tag{20}$$

After all the target vectors for the next iteration have been created, the mutation and recombination processes are repeated. The search for the better solution continues until the maximum number of iterations has been reached and the best updated solution is the solution obtained from DE. The pseudo-code corresponding to the algorithm DE is shown as follows (Algorithm 1):

---

**Algorithm 1** Pseudo-Code of Differential Evolution Algorithm

---

1:   **Set** Iterations, Number of Vectors, $F$, CR
2:     Generate Initial Solution
3:     For i = 1 to Number of Vectors
4:       $X1_{i,1}$ = random number between 0 and 1
5:       $X2_{i,1}$ = random number between 0 and 1
6:       $X3_{i,1}$ = random number between 0 and 1
7:       Target vector solution, calculate objective function and update best solution
8:     **End for**
9:   **For** G = 1 to Max Iteration
10:     Mutation
11:     **For** i = 1 to Number of Vectors
12:       $V1_{i,G} = X1_{best,G} + F(X1_{r1,G} - X1_{r2,G}) + F(X1_{r3,G} - X1_{r4,G})$
13:       $V2_{i,G} = X2_{best,G} + F(X2_{r1,G} - X2_{r2,G}) + F(X2_{r3,G} - X2_{r4,G})$
14:       $V3_{i,G} = X3_{best,G} + F(X3_{r1,G} - X3_{r2,G}) + F(X3_{r3,G} - X3_{r4,G})$
15:       Mutant vector solution, calculate objective function and update best solution
16:     **End for**
17:     Recombination
18:     **For** i = 1 to Number of Vectors
19:       $U1_{ij,G} = \begin{cases} V1_{ij,G}, if(rand(j) \leq CR) \\ X1_{ij,G}, Otherwise \end{cases}$
20:       $U2_{ij,G} = \begin{cases} V2_{ij,G}, if(rand(j) \leq CR) \\ X2_{ij,G}, Otherwise \end{cases}$
21:       $U3_{ijk,G} = \begin{cases} V3_{ijk,G}, if(rand(j,k) \leq CR) \\ X3_{ijk,G}, Otherwise \end{cases}$
22:       Trial vector solution, calculate objective function and update best solution
23:     **End for**
24:     Selection
25:     **For** i = 1 to Number of Vectors
26:       $X_{i,G+1} = \begin{cases} U_{i,G}, if f(U_{i,G}) < f(X_{i,G}) \\ X_{i,G}, Otherwise \end{cases}$
27:     **End for**
28:   **End for**
29:   Return best solution

---

## 5. Results

### 5.1. Solving Problems with Mathematical Models

In the beginning, Lingo version 17 running on an Intel(R) CoreTM i7–8700 CPU E8400 @ 3.20 GHz Ram 16 GHz, was used to solve the problem of the case study with the original mathematical model, which has the non-linear objective function (1) and the non-linear constraint (6). The result from Lingo is shown in Figure 3a. As seen, the local optimal solution is obtained with the objective function value of 657,403 THB/month, compared to 948,470 THB/month, which is the result of Lingo version 11 from [6], and the computational time is 1 h 30 min and 51 s. To achieve the global optimal solution, the non-linear objective function, the non-linear constraint (6), and constraint (8) are transformed to the linear objective function (2) and the linear constraints (7) and (9), respectively. Then, the modified mathematical model is solved using Lingo again. Figure 3b displays the result from Lingo. The global optimal solution is found with the objective function value of 569,563 THB/month, and the computational time reduces to only 1 min and 45 s.

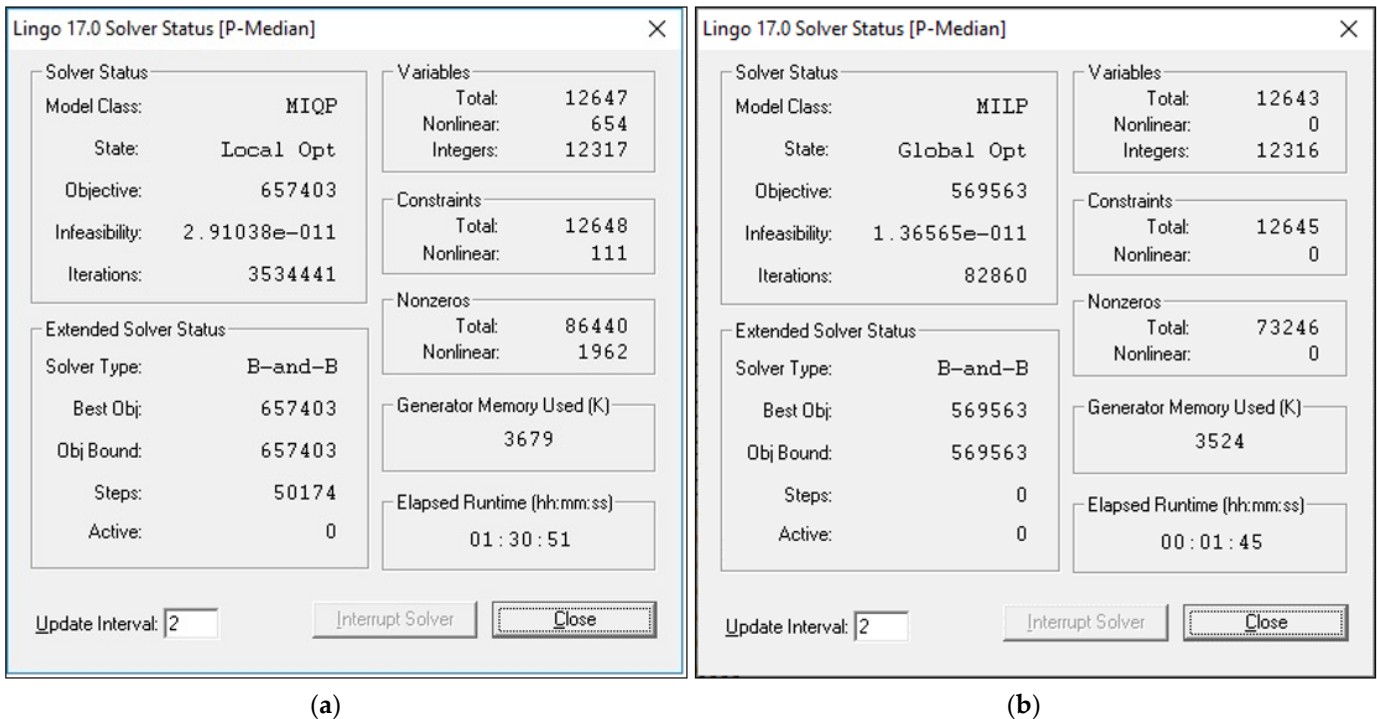

**Figure 3.** Searching for answers to the Lingo program with mathematical equations before (**a**) and after (**b**) development.

Now, the proposed DE algorithm is applied to solve the problems. The algorithm is coded in C++ through Dev C++ version 5.11, running on the same computer used for Lingo. In addition to the case study, three sizes of problems—small (50 community hospitals), medium (100), and large problems (150)—are also tested to measure the performance of the proposed algorithm. All instances have been randomly generated as follows. The average amount of infectious waste (kilogram per month) and the frequency of the collection of the infectious waste per month from each community hospital are randomly generated from the uniform distribution on [80,4000] and [4,8], respectively. The locations of the community hospitals are generated uniformly in the square $[0,1000]^2 \subset R^2$, and Euclidean distances are used to measure the transportation costs. The relevant parameters are defined as shown in Table 5. A preliminary experiment is conducted to evaluate the predefined parameters (F, CR), which are given in Equations (14)–(19). The DOE method of central composite design (CCD) is used to determine the optimal values of F and CR. The corresponding results are analyzed using the Minitab software (Minitab Inc.). The response optimizer results are shown in Figure 4, and therefore, these predefined parameters will be used in later experiments, as follows: F = 1.4818 and CR = 0.2774.

**Table 5.** Relevant parameter configurations for the differential evolution algorithm.

| Parameter | Configure the Relevant Parameters |
| --- | --- |
| 1. Maximum Iterations | 1000 |
| 2. Number of Vectors | Four times the number of hospitals |
| 3. Scaling Factor: *F* | 1.4818 |
| 4. CR | 0.2774 |

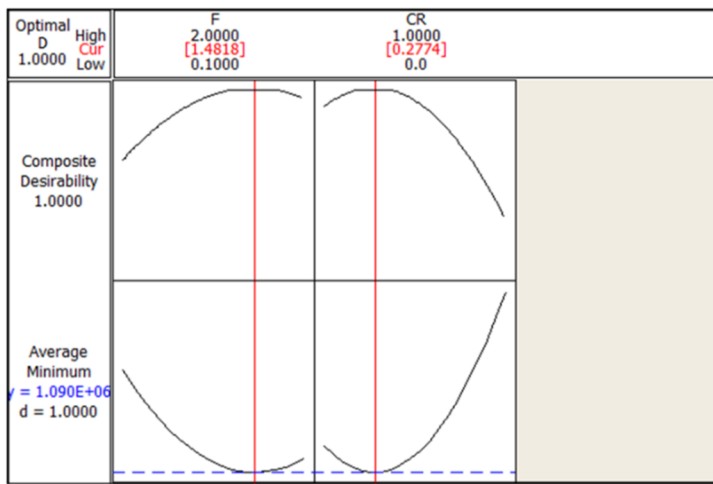

**Figure 4.** Predefined parameter optimization.

For small instances, computational experiments are conducted to measure the performance of DE and Lingo and to demonstrate how well the modified mathematical model works. Both mathematical models have been verified to be accurate. Table 6 displays the computational results. The results indicate that Lingo, with the original mathematical model, can find the solution with the local optimal state. On the other hand, DE and Lingo with the modified mathematical model can find global optimal solutions within a short processing time.

**Table 6.** Computational Results of DE and Lingo for small instance (50 Nodes).

| Instance | Lingo—Original Mathematical Model | | | DE | | Lingo—Modified Mathematical Model | | |
| --- | --- | --- | --- | --- | --- | --- | --- | --- |
| | Status | Total Cost (THB/Month) | Processing Time(s) | Best Total Cost (THB/Month) | Best Processing Time(s) | Status | Total Cost (THB/Month) | Processing Time(s) |
| 1 | Local Optimal | 632,639 | 58 | 610,504 | 17.292 | Global Optimal | 610,504 | 15 |
| 2 | Local Optimal | 742,170.6 | 53 | 699,410.6 | 4.984 | Global Optimal | 699,410.6 | 8 |
| 3 | Local Optimal | 672,856.7 | 55 | 641,236.7 | 16.636 | Global Optimal | 641,236.7 | 7 |
| 4 | Local Optimal | 758,315 | 50 | 723,335 | 4.296 | Global Optimal | 723,335 | 18 |
| 5 | Local Optimal | 746,705 | 60 | 707,319.96 | 1.937 | Global Optimal | 707,319.96 | 25 |
| 6 | Local Optimal | 671,182.9 | 59 | 659,862.9 | 40.827 | Global Optimal | 659,862.9 | 5 |
| 7 | Local Optimal | 636,229.8 | 51 | 595,729.8 | 22.726 | Global Optimal | 595,729.8 | 31 |
| 8 | Local Optimal | 668,438.1 | 63 | 620,997.6 | 14.71 | Global Optimal | 620,997.6 | 21 |
| 9 | Local Optimal | 715,228.6 | 51 | 682,633.6 | 50.261 | Global Optimal | 682,633.6 | 33 |
| 10 | Local Optimal | 719,170.1 | 57 | 617,558.4 | 12.681 | Global Optimal | 617,558.4 | 31 |

Table 7 shows the percentage gap of the total cost and processing time of DE and Lingo with the modified mathematical model compared to the result from Lingo with the original mathematical model. The percentage gap can be calculated by using (21), where $R_v$ is the solution obtained from the proposed DE and Lingo with the modified mathematical model, and $R_L$ is the solution obtained from Lingo with the original mathematical model. As can be seen, for small instances the proposed DE and Lingo with the modified mathematical model outperform Lingo with the original mathematical model, with an average percentage gap of 5.77% for the total cost and around 60% for the computational time.

$$Percentage\ Gap = \left( \frac{R_v - R_L}{R_L} \right) \times \ldots \% \tag{21}$$

**Table 7.** Percentage gap of the total cost and the processing time of DE and Lingo with the modified mathematical model compared to the results of Lingo with the original mathematical model.

| Instance | Total Cost | | Processing Time | |
|---|---|---|---|---|
| | **DE** | **Lingo—Modified Mathematical Model** | **DE** | **Lingo—Modified Mathematical Model** |
| 1 | −3.50 | −3.50 | −70.19 | −74.14 |
| 2 | −5.76 | −5.76 | −90.60 | −84.91 |
| 3 | −4.70 | −4.70 | −69.75 | −87.27 |
| 4 | −4.61 | −4.61 | −91.41 | −64.00 |
| 5 | −5.27 | −5.27 | −96.77 | −58.33 |
| 6 | −1.69 | −1.69 | −30.80 | −91.53 |
| 7 | −6.37 | −6.37 | −55.44 | −39.22 |
| 8 | −7.10 | −7.10 | −76.65 | −66.67 |
| 9 | −4.56 | −4.56 | −1.45 | −35.29 |
| 10 | −14.13 | −14.13 | −77.75 | −45.61 |
| Average | −5.77 | −5.77 | −66.08 | −64.70 |

Because both DE and Lingo with the modified mathematical model provide the optimal solutions, the computational times of both methods are compared using statistical analysis. The paired *t*-test with a 0.05 significance level is conducted to evaluate whether DE and Lingo provide different computational times. Table 8 shows a *p*-value of the test of the hypothesis using a paired *t*-test which is greater than 0.05. Therefore, it can be concluded that the average computational times of DE and Lingo with the modified mathematical model are not significantly different.

**Table 8.** Result of the paired *t*-test for the average computational time of DE and Lingo with modified mathematical model for small instances.

| Detail | *p*-Value |
|---|---|
| Processing Time | 0.895 |

To measure the performance of the developed DE, the results obtained from DE are compared to the ones obtained from Lingo that solve the modified mathematical model of the problem. Table 9 displays the comparison of the results from both methods. It can be seen that both methods can find the global optimal solutions of all 30 instances. For the large size of the problem (instances 21–30), DE provides much less computational time. Finally, for the case study, both Lingo and DE can find the global optimal solution. However, DE outperforms Lingo in the computational time aspect, which is confirmed by the results of the hypothesis testing, as shown in Table 10.

From Table 10, the null hypotheses are rejected for both tests. It can be concluded that the average computational time of DE is significantly less than that of Lingo with the modified mathematical model.

From Figure 5, for small-sized instances, Lingo and DE can find optimal solutions with similar short processing times. However, Lingo takes longer time to find the solutions for the medium- and large-sized instances. The computational time of Lingo increases exponentially as the problem size increases, while that of DE increases linearly. Therefore, it can be concluded that DE outperforms Lingo in terms of the computational time.

**Table 9.** Comparison of results from the Lingo program and the differential evolution algorithm.

| Instance | Number of Hospital | Result from Lingo | | | Result from DE | | Difference in Total Cost (%) |
|---|---|---|---|---|---|---|---|
| | | Status | Total Cost (THB/Month) | Processing Time(s) | Best Total Cost (THB/Month) | Best Processing Time(s) | |
| 1 | 50 | Global optimal | 610,504 | 15 | 610,504 | 17.292 | 0.0000 |
| 2 | 50 | Global optimal | 699,410.6 | 8 | 699,410.6 | 4.984 | 0.0000 |
| 3 | 50 | Global optimal | 641,236.7 | 7 | 641,236.7 | 16.636 | 0.0000 |
| 4 | 50 | Global optimal | 723,335 | 18 | 723,335 | 4.296 | 0.0000 |
| 5 | 50 | Global optimal | 707,319.96 | 25 | 707,319.96 | 1.937 | 0.0000 |
| 6 | 50 | Global optimal | 659,862.9 | 5 | 659,862.9 | 40.827 | 0.0000 |
| 7 | 50 | Global optimal | 595,729.8 | 31 | 595,729.8 | 22.726 | 0.0000 |
| 8 | 50 | Global optimal | 620,997.6 | 21 | 620,997.6 | 14.71 | 0.0000 |
| 9 | 50 | Global optimal | 682,633.6 | 33 | 682,633.6 | 50.261 | 0.0000 |
| 10 | 50 | Global optimal | 617,558.4 | 31 | 617,558.4 | 12.681 | 0.0000 |
| 11 | 100 | Global optimal | 1,078,002 | 124 | 1,078,002 | 392.462 | 0.0000 |
| 12 | 100 | Global optimal | 1,095,953 | 3,239 | 1,095,953 | 412.25 | 0.0000 |
| 13 | 100 | Global optimal | 1,070,054 | 143 | 1,070,054 | 299.107 | 0.0000 |
| 14 | 100 | Global optimal | 1,108,291 | 161 | 1,108,291 | 554.625 | 0.0000 |
| 15 | 100 | Global optimal | 1,074,216.07 | 1,986 | 1,074,216.07 | 761.654 | 0.0000 |
| 16 | 100 | Global optimal | 1,050,140 | 2,210 | 1,050,140 | 558.452 | 0.0000 |
| 17 | 100 | Global optimal | 1,081,190 | 5,033 | 1,081,190 | 307.007 | 0.0000 |
| 18 | 100 | Global optimal | 1,112,685 | 5,420 | 1,112,685 | 428.688 | 0.0000 |
| 19 | 100 | Global optimal | 1,106,656 | 1,900 | 1,106,656 | 401.476 | 0.0000 |
| 20 | 100 | Global optimal | 1,071,397 | 3,637 | 1,071,397 | 283.793 | 0.0000 |
| 21 | 150 | Global optimal | 1,469,615 | 16,556 | 1,469,615 | 2,315.933 | 0.0000 |
| 22 | 150 | Global optimal | 1,467,808 | 27,258 | 1,467,808 | 1505.632 | 0.0000 |
| 23 | 150 | Global optimal | 1,526,718 | 38,603 | 1,526,718 | 1674.715 | 0.0000 |
| 24 | 150 | Global optimal | 1,535,098 | 40,011 | 1,535,098 | 2,031.46 | 0.0000 |
| 25 | 150 | Global optimal | 1,432,549 | 42,911 | 1,432,549 | 2,272.94 | 0.0000 |
| 26 | 150 | Global optimal | 1,511,526 | 148,775 | 1,511,526 | 1672.97 | 0.0000 |
| 27 | 150 | Global optimal | 1,508,135 | 212,837 | 1,508,135 | 2165.418 | 0.0000 |
| 28 | 150 | Global optimal | 1,450,927 | 52,503 | 1,450,927 | 2157.674 | 0.0000 |
| 29 | 150 | Global optimal | 1,444,293 | 128,805 | 1,444,293 | 1552.48 | 0.0000 |
| 30 | 150 | Global optimal | 1,465,037 | 117,992 | 1,465,037 | 2382.795 | 0.0000 |
| Case study | 109 | Global optimal | 569,562.66 | 105 | 569,562.66 | 11.928 | 0.0000 |

**Table 10.** Results of the paired *t*-test for average computational time of DE and Lingo with modified mathematical model for all 30 instances.

| Null Hypotheses | Alternative Hypotheses | *p*-Value |
|---|---|---|
| $H_0: \mu_d = 0$ | $H_1: \mu_d \neq 0$ | 0.008 |
| $H_0: \mu_d \leq 0$ | $H_1: \mu_d > 0$ | 0.004 |

$\mu_1$ = average computational time of Lingo with the modified mathematical model. $\mu_2$ = average computational time of DE. $\mu_d = \mu_1 - \mu_2$.

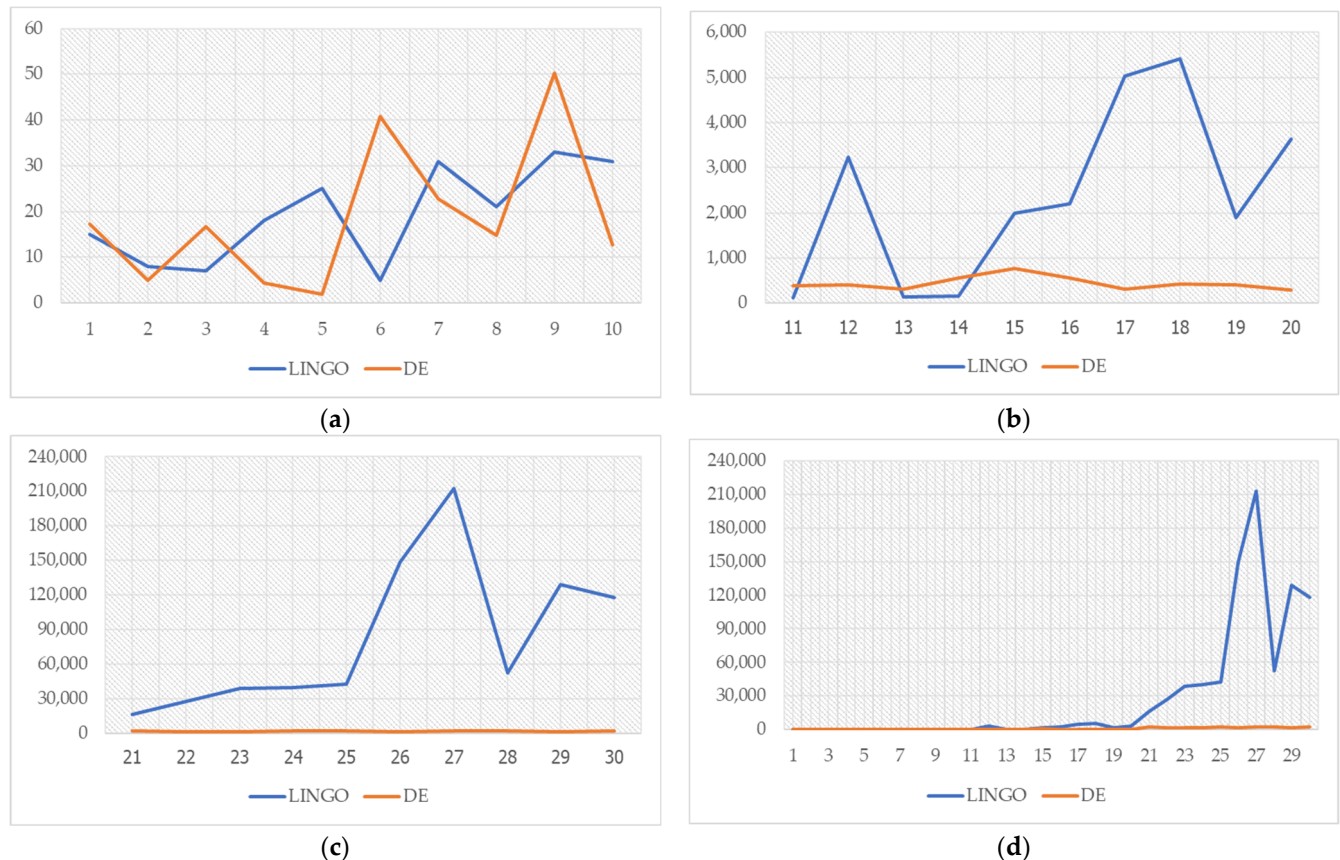

**Figure 5.** Processing time plots of all instances: (**a**) small-sized instances; (**b**) medium-sized instances; (**c**) large-sized instances; and (**d**) all instances.

*5.2. Case Study*

From Table 11, it can be seen that PSO, ILS, and the mathematical programming in the past research have not been able to find the optimal solution. Therefore, in this research the mathematical model was adjusted to become linear, which made it possible to solve to optimality.

**Table 11.** Comparison of the total cost of solving case studies.

| Method | Status | Total Cost (THB/Month) | Difference in Total Cost with Optimal Solution (%) |
|---|---|---|---|
| Original mathematical model by Lingo [6] | Local Optimal | 657,402.66 | 13.36 |
| Particle Swarm Optimization (PSO) [6] | - | 588,298 | 3.18 |
| Iterated Local Search (ILS) [7] | - | 570,183 | 0.11 |
| Mathematical model (Linear) by Lingo | Global Optimal | 569,562.66 | 0.00 |
| Differential Evolution (DE) | - | 569,562.66 | 0.00 |

As mentioned in Section 5.1, DE can solve to optimality the problem of selecting the size and location of the infectious waste incinerators in the upper part of northeastern Thailand. DE can find the global optimal for every repetition and has less computational time than Lingo. The optimal results shown in Table 12 revealed that the total cost is THB 569,562.66, which consists of the cost of transporting the infectious waste of THB 245,400, fixed operating expenses of THB 124,562, and the cost of operating an incinerator for infectious waste of THB 199,600.66. The average time for finding the best solution is 56.80 s, and the total processing time is 154.44 s on average. Two locations, Location 25 (Nonghan Hospital) and Location 52 (Kosumphi-sai Hospital), were selected as the infectious waste

disposal facilities. Both facilities operate the same type of incinerator, with a burning capacity of 300 kg per hour. The facility located at Nonghan Hospital serves 70 hospitals, and the other, located at Kosumphisai Hospital, provides services for 39 hospitals.

**Table 12.** The result of selection of the suitable infectious waste disposal locations and incinerators.

| Suitable disposal Facilities | Hospitals | Type of Incinerator | Burning Time (Hour/Month) | Total Cost (THB/Month) |
|---|---|---|---|---|
| P25 | H1, H2, H3, H4, H7, H18, H20, H21, H22, H23, H24, H25, H26, H27, H28, H29, H30, H31, H32, H33, H34, H35, H36, H37, H38, H39, H40, H41, H42, H43, H44, H45, H46, H47, H48, H49, H50, H68, H69, H73, H74, H75, H76, H77, H78, H79, H80, H81, H82, H83, H84, H85, H86, H87, H88, H89, H90, H92, H94, H95, H96, H97, H102, H103, H104, H105, H106, H107, H108, H109 (70 hospitals) | 300 Kg/Hr. | 209.75 | 569,562.66 |
| P52 | H5, H6, H8, H9, H10, H11, H12, H13, H14, H15, H16, H17, H19, H51, H52, H53, H54, H55, H56, H57, H58, H59, H60, H61, H62, H63, H64, H65, H66, H67, H70, H71, H72, H91, H93, H98, H99, H100, H101 (39 hospitals) | 300 Kg/Hr. | 150.54 | |

## 6. Conclusions and Discussion

Disposing of the infectious waste is a concern for all hospitals. In northeastern Thailand, there are 109 community hospitals, whose monthly total amount of infectious waste is round 104,487 kg. Almost all of them use services of remote private companies to eliminate their infectious waste at a very high cost. The policy of having the infectious waste disposal facilities in the same region as the community hospitals will help to reduce the total system costs, especially the transportation cost. The differential evolution (DE) algorithm is proposed to solve the problems of choosing the suitable location in which to establish the infectious waste disposal facility, the suitable type of the incinerator to use at each facility, and the optimal assignment of the community hospitals to each facility.

In this problem, each community hospital can be a potential location for the infectious waste disposal facility. There are three types of the incinerator. Each facility can use only one incinerator. The transportation of infectious waste from the hospital to the facility is by direct shipping. The non-linear original mathematical model of the problem was transformed to the linear form that can be more easily solved by Lingo. In addition to the case study, thirty random instances were also tested to measure the performance of the proposed DE. For all the instances, both DE and Lingo could find the optimal solutions. In the computational time aspect, DE and Lingo performed equally well for small problem instances, but for the medium- and large-sized problem instances, Lingo had a much longer computational time than DE. Therefore, it can be concluded that DE outperforms Lingo. For the case study, both DE and Lingo can solve the problem to optimality, but for the computational time, DE outperforms Lingo as well. The optimal result is to locate the infectious waste disposal facility at Nonghan Hospital and Kosumphisai Hospital. Each facility utilizes an incinerator with a burning capacity of 300 kg per hour.

From the previous research, PSO [6] and ILS [7] were proposed to solve the case study that was formulated as a non-linear mathematical model. Neither of them could find the

optimal solution. On the other hand, the DE developed in this research could solve the problem to optimality with less computational time.

In the future research, a specific local search will be conducted with DE, and the scope of the problem will be expanded to cover all community hospitals in the northeastern part of Thailand. A hybrid DE will also be developed.

**Author Contributions:** Conceptualization, T.S. (Thitiworada Srisuwandee); methodology, T.S. (Thitiworada Srisuwandee); software, T.S. (Thitiworada Srisuwandee); validation, S.S.; formal analysis, T.S. (Thitiworada Srisuwandee); writing—original draft preparation, T.S. (Thitinon Srisuwandee) and S.S.; writing—review and editing, T.S. (Thitinon Srisuwandee) and S.S.; funding acquisition, S.S. All authors have read and agreed to the published version of the manuscript.

**Funding:** This research received no external funding.

**Data Availability Statement:** Not applicable.

**Conflicts of Interest:** The authors declare no conflict of interest.

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
