# Peer review of "The Differential Evolution Algorithm for Solving the Problem of Size Selection and Location of Infectious Waste Incinerator"

_computation, doi:10.3390/computation11010010_

Round 1

Reviewer 1 Report

Reference to DE is done in the past tense. I think this should be altered in the present tense. Apart from that (which is a minor detail), I think it is more efficient to explain lines 140-147 with maths. Since this is happening in section 4, I don't see why an extended part of section 2 is devoted to the explanation of DE.

No literature review on the methods applied to the problem exists in the manuscript. What approaches are state-of-the-art? What are the limitations these approaches face? The limitations/disadvantages should provide the motivation for this study, i.e., proposing a new method to solve this problem.

The assumptions made to convert the non-linear terms into linear ones seem to simplify the problem. Thus, it doesn't apply to real-world cases. I suggest elaborating more on that matter.

In the experimental section, the computational times of DE are way greater than the ones of mathematical programming. Since both are able to find the global optimum in several cases, then DE does not contribute anything. It's not quicker and it doesn't provide better performance.

Μinor spell check is required since several mistakes occur in the manuscript.

Reviewer 2 Report

A methodology to solve the problem of size selection and location of infectious waste incinerators for community hospitals is presented in this article. The proposed methodology is based on a differential evolution algorithm with the objective of minimizing the total system cost, considering the system cost as the cost of transporting infectious waste, the fixed costs, and the variable cost of operating the infectious waste incinerator.  The novelties and contributions of this work are moderate. However, the topic of the paper is interesting, the scientific soundness and the references used in the manuscript are adequate, the methodology and the results are clearly presented, and the conclusions are supported by the results presented. For these reasons, I recommend accept in present form.

Author Response

Thank you for your comments and suggestions.

Reviewer 3 Report

The submitted manuscript is called “The Differential Evolution Algorithm for Solving the Problem of Size Selection and Location of Infectious Waste Incinerator” and is focused on an interesting issue of the disposal of infectious waste remains. The introduction section looks reasonable, however, the rest of the manuscript requires substantial refinement in order to be accepted for publication.

The description of differential evolution is rather poor: a lot of words are said about basic principles of DE and almost nothing about numerous modifications designed to improve the convergence and effectiveness of DE. Section 4 contains a very long description of four main stages of DE, which everyone knows.

The tests with mathematical models were done for a particular settings of DE (F=2, CR=0.8). Why particular these values? It is known that the convergence of DE is sensitive to the settings of F and CR. The impact of F and CR values needs to be investigated on a task-by-task basis.

Discussion is missing completely! Any scientific paper must have the Discussion section. Comparison the authors’ results with those of obtained before must be done…This is the only the way to get an idea of the work that has been done.

Round 2

Reviewer 1 Report

The authors addressed some of the comments. The updated Related Literature section fixed some issues of the previous version regarding the motivation of the work. Also, the additional text in lines 76-85 provides one more argument. I think the authors should highlight that their main contribution is the new formulation they propose. 

I could not find the references [6] and [7]. So, I cannot verify if the motivation presented by the authors exists.

"The case study is a problem that requires many answers": what do the authors mean by "answers"? Solutions? Decision variables? Clarify that in the text and revise the sentence accordingly.

For instances 1-30, as I mentioned, the computational time of DE is greater than the mathematical programming method. Obviously, the exact method would need more time as the instances grow. But the problem is that for smaller instances, the big difference in processing times does not justify the proposition of the meta-heuristic. If the meta-heuristic needs so much time to find a (near-)optimal solution while MP locates the optimal one quickly, the usage of a meta-heuristic is not necessary.

Furthermore, it's very strange that DE needs only 11 seconds to find the optimal solution for the case study when the MP model needs 105. We would expect the same performance in smaller instances. The codes of DE should be provided in a repository to enable the reproducibility of this work.

Round 3

Reviewer 1 Report

I don't understand why the part "the nature of the problem that Lingo Solver can solve to optimality" is meaningful. In Operations Research, it does not matter if a specific solver is able to find the optimal solution. It would be more meaningful to mention that the formulation has been validated and stands.

Even though reference 7 was altered, references 6 and 7 do not exist. I searched them and no scholar database can retrieve them.

Line 467: "... Lingo with the original mathematical model cannot find an optimal solution" - This sounds very strange. I am not sure what the authors mean. Obviously, solving the problem according to a different formulation means that you solve a new problem. The optimal solution for each problem (i.e., formulation) is different.
